# Chemometric Classification of Apple Cultivars Based on Physicochemical Properties: Raw Material Selection for Processing Applications

**DOI:** 10.3390/foods12163095

**Published:** 2023-08-17

**Authors:** Maiqi Zhang, Yihao Yin, Yantong Li, Yongli Jiang, Xiaosong Hu, Junjie Yi

**Affiliations:** 1Faculty of Food Science and Engineering, Kunming University of Science and Technology, Kunming 650500, China; zhangmaiqi0518@163.com (M.Z.); iscoyyh@hotmail.com (Y.Y.); lyt190418@163.com (Y.L.); yongli_jiang0617@163.com (Y.J.); huxiaos@263.net (X.H.); 2Yunnan Engineering Research Center for Fruit & Vegetable Products, Kunming 650500, China; 3International Green Food Processing Research and Development Center of Kunming City, Kunming 650500, China; 4College of Food Science and Nutritional Engineering, China Agricultural University, Beijing 100083, China

**Keywords:** apple cultivars, classification, apple production, characteristic qualities, chemometrics

## Abstract

Apple cultivars exhibit significant diversity in fruit quality traits, creating distinct consumption scenarios. This study aimed to assess the physicochemical parameters and sensory attributes differences among fifteen apple cultivars and identify characteristic qualities suitable for various processed apple products using chemometric analysis. Relatively large differences were registered between cultivars for deflection, peel color, titratable acidity (TA), the ratio of total soluble solid to titratable acidity (TSS/TA), hardness, soluble sugar, and volatile organic compound contents. Sensory results showed significant differences existed among the preferences for different processed products. Based on the above results, all cultivars could be distinguished into three main clusters. Cluster I (i.e., Aziteke, Bakeai, Magic Flute, Royal Gala, Red General, Red Delicious, and Zhongqiuwang) demonstrated favorable appearance, high sensory scores, and rich aroma volatile compounds, making them suitable for direct consumption. Cluster II (i.e., Fuburuisi, Sinike, Honglu, and Huashuo) exhibited a higher sugar and acid content, making them suitable for apple juice production. Cluster III (i.e., Miqila, Honey Crisp, Shandong Fuji, and Yanfu 3) were more suitable for fresh-cut apples due to their good flavor and undesirable appearance. Several chemometric analyses effectively assessed differences among apple cultivars.

## 1. Introduction

Apple (*Malus* × *domestica* Borkh.) is a widely cultivated and consumed fruit globally, known for its distinctive flavor and widely appreciated by consumers [1]. China, with the largest apple planting area and the highest total production in the world, has a significant contribution to the global apple market. It was reported that apple production in China was 40,501,041 tons in 2020, accounting for 46.85% of the world’s apple production [2]. Despite the majority of apples being consumed as fresh fruit, the increase in consumer demand for natural and minimally processed fruit products has led to a growing interest in apple products [3]. According to the World Health Organization (WHO), the recommended daily intake of fruits and vegetables is more than 400 g per capita [4]. Fresh-cut apples have gained popularity as a convenient and nutritious snack, particularly in school lunch programs and for family consumption. This segment of the market is expected to continue growing as it offers convenience while contributing to daily fruit consumption. According to statistics from the United States Department of Agriculture (USDA), fresh apple slices accounted for approximately 1.4% of the total US apple crop [5]. Apple juice (freshly pressed cloudy apple juice), also a minimally processed product, is growing in market demand and value because of its freshness, convenience, nutritional value, and health benefits [6].

The quality of raw materials is a crucial factor in determining the overall quality of apples and apple products, which has been shown to be influenced by a variety of factors, including environmental factors, storage time, fruit maturity, post-harvest treatment, cultivar, and processing [7,8]. In general, phenotypes (size, shape, and color), flavor (aroma, taste, and mouthfeel sensations), and nutrients are the main quality determining factors of apples [9,10]. Among the various factors, the flavor is considered to be the primary determinant of apple consumption, which is influenced by a complex interplay of organic acids, sugars, volatile compounds, and texture [11]. According to the commercial standards for apple quality in Europe, the USA, and China [9], appearance (e.g., integrity, smooth peel, and absence of russeting) affects consumer preferences for apples consumed directly. However, when evaluating fresh-cut apples, freshness is often associated with mouthfeel sensation and taste [12], as differences in appearance are eliminated during the cutting process. Furthermore, evidence from previous studies has indicated that sensory attributes related to a “Fresh-like” perception play a significant role in consumer preference for apple juice [13]. Even though numerous studies have focused on the evaluation of apple quality at the cultivar level [14,15], the specific sensory characteristics suitable for different processed apple products have yet to be thoroughly investigated.

Therefore, the objective of this study was to comparatively evaluate the sensorial quality of fifteen apple cultivars and to deeply investigate the characteristic quality attributes suitable for different processed apple products’ needs. To achieve the objectives, a wide range of quality attributes was analyzed, including fruit phenotypes, taste attributes, aroma profiles, and visual appearance. In addition, the apples were prepared with whole apples, fresh-cut apples, and apple juice for sensory evaluation. Considering the large data sets obtained in this study, correlation analysis between apple products and sensorial quality attributes was conducted using multi-chemometric approaches. The outcome of this study may offer valuable insights for selecting apple cultivars to meet diverse processing requirements.

## 2. Materials and Methods

### 2.1. Reagents and Chemicals

Standards for high-performance liquid chromatography (HPLC) were purchased from Shanghai Yuanye Bio-Technology Co., Ltd. (Shanghai, China) and included glucose, fructose, sucrose, tartaric acid, malic acid, citric acid, and quinic acid. The HPLC-grade reagents methanol and acetonitrile were acquired from Sigma-Aldrich Chemical Co., Ltd. in Shanghai, China. All additional chemicals were purchased from Sinopharm Chemical Reagent Co., Ltd. in Shanghai, China, and were of analytical quality.

### 2.2. Plant Materials

Fifteen apple cultivars were obtained from Zhaotong City in Yunnan province, China. The apples were randomly sampled at full ripening in 2021 from the same orchard owned by Zhaotong Transcendental Agriculture Co., Ltd., Shanghai, China. The maturity of the apple fruits was determined by the company based on fruit color, starch index, and days after pollination. A total of 150 kg of fruits (10 kg per cultivar) were swiftly transported to the laboratory. After cleaning, the fresh apple samples underwent a series of fruit quality analyses, including measurements of length, width, fruit shape index, fruit deflection index, color, hardness, and chewiness.

To conduct further analysis, samples from each of the fifteen apple cultivars were randomly selected (Twenty fruits per cultivar) and processed by removing the stalks, sepals, and cores. The apples were then cut into small pieces and pulped using a wall breaker. The obtained apple pulp was immediately frozen in liquid nitrogen and stored at −80 °C for subsequent analysis of pH, titratable acidity (TA), total soluble solids (TSS), sugar, organic acid profile, electronic nose (E-nose), and headspace-solid phase microextraction-gas chromatography-mass spectrometry (HS-SPME-GC-MS). All analytical procedures were performed in three biological replicates to ensure the accuracy and reproducibility of the results.

For further sensory evaluation, apples were washed with distilled water and randomly divided into three parts. The first part was prepared for whole fruit consumption without any other treatments. The second part of apples was cored and squeezed using a juicer (JYZ-E16, Joyoung Co., Ltd., Hangzhou, China) to obtain fresh cloudy apple juice. The rest of the apples were cored and cut into slices to obtain fresh-cut apples.

### 2.3. Fruit Phenotypes

Length and width were measured by a vernier caliper (minimum measure 0.01 mm). The shape index was calculated by the ratio of fruit length to diameter. The deflection index was analyzed according to the following Equation (1). All measurements were carried out six times.
(1)$$\mathrm{DD}=\frac{2\left(R\times H-r\times h\right)}{\left(R\times H+r\times h\right)}\;$$
where *H* is the height of the large fruit surface (mm), *R* is the distance from the large fruit surface to the center of the fruit (mm), *h* is the height of the small fruit surface (mm), *r* is the distance from the small fruit surface to the center of the fruit (mm).

### 2.4. Fruit Color

For each cultivar, six apples were randomly chosen to evaluate both the peel and flesh color. Four peel samples were taken symmetrically along the equator of each apple, and the flesh was cut into small cubes (3 × 3 × 3 cm^3^) from the same location. A colorimeter (Agera, Hunter Associate Laboratory, Inc., Fairfax, VA, USA) was used with illuminant D65 and 10° observer angle, providing color space coordinates (*L**, *a**, and *b**). In addition, the Colortell color tool (https://www.colortell.com/ (accessed on 1 November 2021) was used to fit colors according to the color parameters. Each peel and cube from a specific apple underwent color analysis individually. In total, the color of the peel or flesh for each apple cultivar was determined in 24 separate instances.

### 2.5. pH, TSS, and TA Determination

The apple pulp was centrifuged at 10,000× *g* for 15 min at 4 °C, and the supernatant was used for measurements of pH and TSS with a pH meter (FE28-Standard, Mettler Toledo, Zurich) and a refractometer (TD-45, Jinkelida, Beijing, China) at 20 °C, respectively. TA was analyzed using an automatic potentiometric titrator (907 GPD Titrino, Metrohm, Herisau, Switzerland) according to Equation (2) [16]. All measurements were carried out in triplicate.
(2)$$\mathrm{TA}\left(\%\right)=\frac{C\;\times \;V\;\times \;K}{m}\;\times \;100\;$$
where *C* is the NaOH concentration (0.1 mol/L), *V* is the NaOH volume used (mL), *m* is the weight of apples pulp (g), and *K* is the citric acid conversion factor (0.064).

### 2.6. Hardness and Chewiness Determination

The hardness and chewiness of apple samples with peel were measured using a Texture Analyzer (TA-XT. Plus, Stable Micro Systems Ltd., Godalming, UK) equipped with a P/50 probe (diameter of 50 mm). Six samples of each cultivar were used in this experiment. Four small cubes (2 × 2 × 2 cm^3^) were symmetrically prepared from the flesh samples along the equator of each apple, ensuring they were as far away from the apple stem as possible. These cubes were then subjected to a two-cycle compression. The texture parameters were set as follows: a test speed of 2 mm/s, a measurement distance of 10 mm, a compression time of 5 s, and a trigger force of 5 N.

The following texture parameters were measured:

Hardness (N): Maximum force observed during the first compression cycle;

Cohesiveness: Ratio of the total energy required for two compressions;

Springiness: Rate at which a deformed material returns to its unreformed condition after removing the deforming force;

Chewiness (N): Work needed to masticate the sample before swallowing, calculated as the multiplication of hardness, cohesiveness, and springiness.

All data obtained were analyzed using the built-in software of the texture analyzer. Each cube from a specific apple was compressed individually. A total of 24 measurements were conducted to determine the hardness and chewiness of each apple cultivar.

### 2.7. Sugar and Organic Acid Profile Determination

The sugar profile analysis was conducted following the method of Yi et al. [17] with slight modifications. The assay was performed in triplicate. Initially, randomly selected samples were cut into small pieces after removing the stalks, sepals, and cores, and pulped using a wall breaker. Then, 1 g of the sample was mixed with 5 mL milli-Q water, followed by the addition of 200 μL of each K_4_[Fe(CN)_6_] (15% *w*/*v*) and ZnSO_4_ (30% *w*/*v*) were added. After resting for 30 min, the mixture was centrifuged at 10,000× *g* for 15 min at 4 °C. The supernatant was diluted 10 times with milli-Q water and analyzed using a high-performance liquid chromatograph (HPLC) system coupled with evaporative light scattering detection (1260 Series, Agilent Technologies, Santa Clara, CA, USA). Separation of sugar extract (5 μL) was carried out on a column (250 mm × 4.6 mm, 5 μm particle size, Asahipak NH2P-50 4E, Showa Denko KK, Tokyo, Japan) at a flow rate of 1 mL/min using isocratic elution (75% (*v*/*v*) acetonitrile/water) at 30 °C. Sugars were identified and quantified based on retention times and calibration curves of glucose and fructose standard solutions.

The extraction procedure of organic acids was in accordance with a previously described method. Thirty microliters of the extract were analyzed using an HPLC system equipped with a Prevail Organic Acid column (250 mm × 4.6 mm, 5 μm particle size, Alltech Grace, Deerfield, FL, USA) protected with a guard cartridge (7.5 mm × 4.6 mm, 5 μm particle size, Alltech Grace, Deerfield, FL, USA). Separation occurred at 25 °C by isocratic elution (25 mM potassium dihydrogen phosphate buffer pH 2.5) at a flow rate of 0.8 mL/min. A UV-DAD detector at 210 nm was used for detection. Identification and quantification of organic acids were performed based on retention times and calibration curves of standard solutions.

### 2.8. Aroma Profile Determination

#### 2.8.1. E-Nose Analysis

Apple samples were analyzed using a portable universal cNose (Baosheng Industrial Development Co., Ltd., Shanghai, China). The assay was carried out in triplicate. The system comprised a sample device, a detector unit with an array of 18 distinct metal oxide sensors, and corresponding software program for data collection and analysis. Shortly after 3.0 g of samples were put into 10 mL headspace sample vials, the vials were heated in a water bath at 25 °C for 30 min. The E-nose system’s settings were set at 300 mL/min for the chamber flow rate, 100 mL/min for the injection flow rate, and 120 s for measurement. 

#### 2.8.2. HS-SPME-GC-MS Analysis

HS-SPME-GC-MS analysis was performed to measure the volatile compounds in the apple pulp samples. After removing the stalks, sepals, and cores, three fruits were selected, and their pulps were cut into small pieces (3 mm^3^) and mixed. Then, a 20 mL glass container was filled with the sample and weighed with 50 μL of an internal standard mix (50 μL/L of 2-nonanol) and 3 mL of saturated sodium chloride solution. The volatile compounds were extracted using HS-SPME and analyzed using GC-MS at the Kunming University of Science and Technology Analysis and Testing Center (Kunming City, Yunnan province, China).

Volatiles were identified by matching the experimental mass spectra with the standard spectra stored in NIST14 library data (a threshold match of 80%). Alkane mixture (C5–C25) was directly injected into GC-MS under the same operating conditions to calculate the retention index (RI). Internal standard (2-nonanol) calibration was also conducted for semi-quantification.

### 2.9. Sensory Evaluation

Apples were sorted based on their length and width into two categories: small apples and big apples. This classification followed the commercial size grades for apples [18] and took into account diverse consumer preferences for apple size. Small apples were defined as those with both length and width measuring less than 80 mm, while the remaining apples were classified as big apples. To ensure balanced representation among the different sizes, a randomized complete and balanced incomplete block design was employed. For the complete block design, a total of 30 fruits from 5 different cultivars were randomly selected and evaluated by 16 untrained panelists. For the incomplete block design, 60 fruits from 10 varieties were randomly selected and evaluated by 18 panelists with 5 varieties evaluated per panelist. Following Aubert et al. [19], a ranking test (one being the lowest, five being the highest ranked) was carried out for each cultivar to compare quality characteristics (shape, color, aroma, crispness, mealiness, sweetness, sourness, taste, and general evaluation) of the samples. In a random monadic order, three processed apple products, whole fruit, fresh-cut apple, and apple juice, placed in small white plates or transparent vials (10 mL) using three-digit random numbers, were presented. The sensory evaluations were performed in a sensory laboratory using white light and single booths and the data were expressed as the sum of measurement rank scores.

### 2.10. Data Analysis

The mean and standard deviation of each set of data were calculated using SPSS 20.0 statistics software (IBM, Armonk, NY, USA). One-way analysis of variance (ANOVA) was used, and significance was determined at *p* < 0.05 for the Tukey’s significant difference test that followed. Using OriginPro software (version 8, Origin Lab Corporation, Northampton, MA, USA), principal component analysis (PCA), hierarchical cluster analysis (HCA), and correlation analysis were carried out. SIMCA-P 14.1 (Umetrics AB, Umea, Sweden) was used to conduct the partial least squares-discriminant analysis (PLS-DA).

## 3. Results and Discussion

### 3.1. Apple Appearance

#### 3.1.1. Size

Appearance was one of the decisive factors, affecting consumer consumption of apples, such as size, shape, defect-free status, and color [9]. Table 1 shows the length, width, shape index, and deflection index of fifteen apple cultivars. The average length values of apples ranged from 60.33 mm to 83.72 mm, which was in agreement with the observation by Zhao et al. [20]. Among different apple cultivars, the Zhongqiuwang apple had the highest length (83.72 mm), the Honey Crisp apple had the highest width (87.93 mm), but the Sinike apple had the lowest length (60.33 mm) and width (62.53 mm). According to the size, the apple cultivars could be divided into two groups: the big apples (Aziteke, Fuburuisi, Red General, Honglu, Red Delicious, Huashuo, Honey Crisp, Shandong Fuji, Yanfu 3, and Zhongqiuwang) and small apples (Bakeai, Royal Gala, Miqila, Magic Flute, and Sinike). Nowadays, whether it is a big or a small apple, there are different consumer demands [21]. At the same price, the apple with larger sizes might be preferred over the smaller ones [22]. However, consumers also appreciate small apples that can be eaten easily, as they align with the goal of minimizing waste [9].

#### 3.1.2. Shape and Deflection

In addition to size, the shape index reflects the symmetry of the fruit, while the deflection index measures its flatness [23]. As shown in Table 1, all the examined apple cultivars had shape indexes above 0.82, meeting the market standard for minimal misshapen fruits [9]. Most apples showed shape indexes around 0.95, indicating slight oblate spherical shapes. Among different apple cultivars, the Huashuo and Miqila apples had shape indexes close to 1, indicating that their fruits were globe-shaped. The globe-shaped apple often commands higher commercial prices [12,24]. In addition, the highest shape index values were observed on Red Delicious and Magic Flute apples with 1.07, indicating that they were elongated. Honey Crisp and Shandong Fuji apples displayed the lowest shape indexes, showing morphologically oblates shapes.

Table 1 also demonstrates significant variation in the deflection index, ranging from 0.09 to 0.34. The average deflection index of the fifteen apple cultivars was 0.17, with a coefficient of variation of 41.34%. Cultivars such as Honglu, Honey Crisp, Shandong Fuji, Sinike, and Fuburuisi apple displayed higher deflection index values than others, suggesting asymmetrical growth and fruit skewness [23]. This is also confirmed by the visual observation (Figure 1). Many factors influence lopsided apple fruits, such as the state of fruits borne, the number of keeping inflorescence, the nutrients of tree body storing, and light conditions. Liu et al. [23] observed that the number of seeds was reduced and the rate of asymmetrical fruit production was increased when the ‘Fuji’ apple was not adequately pollinated. The symmetry of fruit might be a crucial factor influencing consumer purchasing decisions [25].

#### 3.1.3. Color

The color differences among fifteen apple cultivars were assessed by measuring color values, color phenotype, and fitting graphs. The lower *L** value indicates a darker color and lower brightness, while the a* and b* values provide a description of green-red and blue-yellow colors, respectively [6]. As shown in Table 1, most apple peels exhibited a bright red color (*L** > 0, *a** > 0, *b** > 0). According to the peel color, the apple could be divided into three groups: darker red apples (*L** < 20, Bakeai, Red Delicious, Huashuo, and Magic Flute), middle red apples (20 < *L** < 40, Aziteke, Fuburuisi, Red General, Royal Gala, Honey Crisp, Shandong Fuji, Sinike, and Yanfu 3and brighter red apples (*L** > 40, Honglu, Miqila, and Zhongqiuwang).

No significant difference (*p* > 0.05) in color values was found for apple flesh, including *L** values (75.47–80.02), *a** values (−1.45–3.93), and *b** values (19.54–26.58). It indicated that apple flesh generally exhibits a clean and pale-yellow color. The color value results align with the color phenotype and fitting graph of apple flesh.

### 3.2. Internal Quality

#### 3.2.1. Sweetness and Acidity

The TSS and sugar content of fifteen apple cultivars are shown in Figure 2A,B. The TSS of apples is a key factor reflecting the sweetness of the fruit, as a higher TSS value indicates sweeter fruit [26]. As can be seen, the Aziteke apple had the highest TSS value (16.13 ^◦^Brix), followed by the Shandong Fuji apple (15.60 ^◦^Brix), while Miqila had the lowest TSS value (11.35 ^◦^Brix). When an apple with a TSS value above 12 ^◦^Brix, it was considered to be high-quality fresh apples [14]. There were three apples with TSS values less than 12, including Miqila, Huashuo, and Royal Gala apples. It seemed that most of the Zhaotong apples were almost sweet, especially the Aziteke and Shandong Fuji apples. The sweetness of apples was highly related to their sugar profile and contents. As shown in Figure 2B, fructose (13.85 mg/g to 57.93 mg/g) was dominant, followed by sucrose (4.76 mg/g to 28.58 mg/g) and glucose (5.54 mg/g to 12.87 mg/g). The contents of sugars measured in this study were slightly lower than those measured by Aprea et al. [27]. This inconsistency might be due to the different sampling approaches. Generally, central flesh tastes sweeter and thus middle part (around core) of the flesh had a higher percentage of sugars than the whole apple.

On the other hand, sourness is another taste factor that serves as a key indicator in the industry. The sourness of apple fruit was mainly associated with pH, TA, and content of organic acids, as depicted in Figure 2C–F. The pH levels of apples varied, ranging from 3.54 (Zhongqiuwang) to 4.29 (Honglu) and TA values ranged from 0.19% (Honglu) to 0.42% (Zhongqiuwang). The results were consistent with previous research reported by Guo et al. [28]. According to Harker et al. [10], the lowest perceivable concentration of TA is 0.08%, indicating that all apples in this study had a distinct sour taste. This sourness can be attributed to the presence of organic acids [9]. Among organic acids, citric acid and malic acid were dominant in all samples, with concentrations ranging from 0.07 mg/g to 0.33 mg/g and from 2.48 mg/g to 4.98 mg/g, respectively (Appendix A). Malic acid was found to be the most relevant compound, accounting for 90% of organic acids in apples, which was in accordance with Bai et al. [29].

Neither the sugar content nor the acid content directly reflected the taste of the apple fruit. While the sugar/acid ratio (i.e., TSS/TA ratios) of apples is considered to be an effective parameter reflecting fruit taste harmony [30]. A large range of TSS/TA ratios was observed on different apple cultivars, from 38.21 (Miqila) to 70.71(Honglu), with means of 47.75 and CV of 35.18%. Furthermore, the TSS/TA value is an important factor in determining whether the apple fruit is suitable for direct consumption or processing. The apple cultivars with TSS/TA values between 40 and 50 are recommended for direct consumption [31]. It seems that Aziteke, Bakeai, Honey Crisp, Royal Gala, and Magic Flute apple are suitable for direct consumption. On the other hand, apple cultivars with low TSS/TA ratios may be more suitable for apple juice production [32]. Therefore, Sinike, Fuburuisi, and Zhongqiuwang apple could be considered the ideal cultivars for apple juice production due to their low TSS/TA ratios (<40). The flavor of the apple juice produced by these fifteen apple cultivars was evaluated through sensory evaluation (Section 3.4).

#### 3.2.2. Hardness and Chewiness

Texture profiles can be used to assess fruit sensory quality, with hardness reflecting the compactness and firmness of the fruit and chewiness reflecting the delicacy of mouthfeel [33]. As shown in Figure 2G, H, hardness and chewiness of apple cultivars, fell in the range of 4.12–8.29 N and 0.38–0.95 N, respectively. They were consistent with previous research by Ebadi et al. [34] and Mureșan et al. [32]. Among the apples, Fuburuisi, Sinike, and Aziteke apples belonged to the high-hardness cultivars, indicating that they were almost with high crispness. Firm fruits have denser tissues (smaller cells with less interspace) and can be stored for longer than soft fruits [9,22]. Moreover, firmness as well as the absence of mealiness are the most preferred textural traits by consumers [35].

#### 3.2.3. Apple Aroma

The aroma profile of different apple fruits was fingerprinted using both electronic nose (E-nose) and HS-SPME-GC-MS techniques. The E-nose is capable of detecting the mixture’s overall olfactory impression without separating volatiles [36]. The HS-SPME-GC-MS technique was applied to profiling volatiles in plant and/or food products [15,37]. The results of E-nose and HS-SPME-GC-MS were illustrated in Figure 3A,B and Figure 3C,D respectively.

The radar graph of the E-nose profile (Figure 3A) showed that fifteen apple cultivars exhibited differences in sensor 1, sensor 4, sensor 5, sensor 6, sensor 9, sensor 14, sensor 15, sensor 16, sensor 17, and sensor 18. The results indicated that the profiles of all samples were similar in terms of sensors, but their ranges differed. It was speculated that the odor difference of apples from different producing areas was mainly manifested in the differences in volatile substances such as esters, aldehydes, and alcohols. The response signals of E-nose sensors for the apple sample were subjected to principal component analysis (PCA), with PC1 and PC2 contributing 95.4% of the total variance (Figure 3B). As shown, Sinike and Bakeai apples were distinguished from other cultivars. Aziteke, Miqila, Huashuo, and Yanfu 3 clustered together, implying their similarity in the composition of volatile compounds to some extent.

In this study, a total of 56 volatile compounds were identified and quantified in all apples by HS-SPME-GC–MS, including 24 esters (the most abundant volatiles), 12 aldehydes, 8 alcohols, 3 ketones, and 9 other compounds (Table 2). The esters, aldehydes, and alcohols were the main compounds in apples, followed by ketones and others, which were in agreement with previous studies [14,15,38]. A wide range of volatile compounds was identified, including ethyl butyrate, ethyl-2-methyl butyrate, ethyl butyrate, hexyl acetate, butyl hexanoate, and hexyl hexanoate (Appendix A). These esters have been recognized as the major contributors to the fruity and apple-like aroma profile of apples [36,39,40]. In addition, certain compounds with off-flavors were also detected. Some compounds with off-flavor were also identified. For instance, 1-hexanol exhibited an earthy odor, methyl heptenone had a musty odor, and benzaldehyde presented an odor reminiscent of bitter almond [15]. The number and content of aroma substances varied among apple cultivars, with Zhongqiuwang having the highest number of volatile compounds (33) and Bakeai having the lowest (24). Royal Gala had the highest content of volatile compounds, followed by Magic Flute (117,278.20 μg/kg) and Bakeai (115,658.21 μg/kg), while Aziteke had the lowest volatile compounds content (8909.18 μg/kg). These were the cultivars that were likely to be popular with consumers as aroma was a major factor in their acceptance.

To better understand the changing tendencies of these volatile compounds, their relative abundances in the fifteen apple cultivars were investigated via a hierarchical cluster analysis (Figure 3D). Most of the cultivars contained a large proportion of volatile compounds of esters (comprising over 50% of the total volatile compounds), which contributed to the fresh and fruity apple fragrance [41]. Butyl acetate and ethyl 2-methyl butanoate were the main esters identified, with Royal Gala (30,673.95 μg/kg) and Bakeai (28,240.66 μg/kg) having relatively higher concentrations of butyl acetate, and Yanfu 3 (20,120.95 μg/kg), Fuburuisi (19,500.51 μg/kg), and Shandong Fuji (17,301.72 μg/kg) having more abundant ethyl 2-methyl butanoate. The two esters were liable to contribute significantly to the apple’s typical odor [36]. (E)-2-hexenal and hexanal were the most abundant aldehydes, with green-grassy odor notes [39]. (E)-2-hexenal and hexanal were detected in all cultivars, but the content varied greatly among the apple cultivars, ranging from 1946.63 and 1674.77 μg/kg in Aziteke to 23,696.16 and 33,841.76 μg/kg in Magic Flute. The most dominant alcohol identified was 1-hexanol. As indicated by Guo et al. [28], 1-hexanol had an unpleasant and earthy odor, which may negatively impact the apple fragrance in some cultivars (e.g., Huashuo, Honey Crisp, Red Delicious, Red General, and Bakeai). Ketones are known to have a floral and fruity sweet flavor [42], but they were found in much lower quantities compared to other compounds, and there was little variation in content among different cultivars. In particular, the volatile compounds in Red General and Red Delicious had similar accumulation trends, with higher ester contents, such as propyl acetate, isoamyl hexanoate, butyl propionate, and propyl butyrate. These esters contribute to celery, raspberry, earthy, rose, apricot, and fruity sensorial attributes in apple flavor (Table 2). This finding confirms that the volatile compound profile is highly cultivar-dependent, mainly due to the variation in esters. The content of total volatile compounds and esters was potentially a critical factor in the eventual consumer selection decision.

### 3.3. Sensory Evaluation

The results of the sensory evaluation are presented in Figure 4. Based on consumer consumption habits and needs, the apples were categorized into two groups, small-size cultivars (Bakeai, Royal Gala, Miqila, Magic Flute, and Sinike) and large-size cultivars (Aziteke, Fuburuisi, Red General, Honglu, Red Delicious, Huashuo, Honey Crisp, Shandong Fuji, Yanfu 3, and Zhongqiuwang), and then subjected to the sensory evaluation. For the whole apples, Bakeai, Royal Gala, and Magic Flute received relatively higher scores in shape and color, showing a similar trend of general evaluation. Among large-size cultivars, Aziteke, Red General, Red Delicious, and Zhongqiuwang also scored highly in the same sensory attributes. For fresh-cut apples, Royal Gala and Magic Flute had the highest comprehensive preference degree (18.75 points and 19.38 points) among small-size cultivars, presenting stronger aroma and sweetness. Whereas Yanfu 3, Fuburuisi, and Shandong Fuji were the most preferred cultivars among big-size cultivars. Such high sensory scores of these cultivars could be attributed to the combination of aroma, taste, and mouthfeel sensations. Certain correlations were found between the physicochemical parameters we measured and the sensory attributes. The score tendency of general evaluation and mealiness was the quite opposite, which was in accordance with those of other researchers and shows that assessors preferred fruit with higher firmness [43]. After juicing the apples, Sinike, Fuburuisi, Honglu, and Huashuo presented higher scores in general evaluation and taste, but no consistent relation between color and aroma was observed, indicating that taste attribute in apple juice was significantly associated with changes in consumers’ liking [13].

### 3.4. Chemometrics Analysis

In the present study, PLS-DA and correlation analyses were used to better study the relationships between quality parameters, sensory profiles, and their impact on the differences in apple cultivars. This has been effective in other studies for the comprehensive evaluation of apple quality [14,44]. The general evaluation, shape, and color scores of whole fruit scores in sensory evaluation, as well as total volatile compounds content, were found to cluster together (Figure 5A). It indicated that color and shape were important factors influencing consumer acceptance of apples, with a preference for cultivars exhibiting bright red colors and regular shapes. Notably, in PC2, Honglu apple was located on the negative side, which was highly correlated with *L**, *b**, and deflection, as well as taste scores of apple juice. In contrast, Aziteke, Fuburuisi, Shandong Fuji, and Yanfu 3 apples were strongly associated with most of the sensory attributes, indicating that these cultivars had favorable flavor characteristics. The result was consistent with the quality indexes and sensory evaluation we measured in this study.

As illustrated in Figure 5B, the Pearson significant correlation analysis showed that general evaluations were positively correlated with color (r = 0.81) and shape (r = 0.53), sourness (r = 0.58), and taste (r = 0.80) in sensory evaluation. From these results, it can be speculated that the desirables of evaluators were color and shape, sourness, and taste for whole apples, fresh-cut apples, and apple juice, respectively. In addition, the negative correlations were obtained for the deflection index with W-Shape (shape score in whole apple), W-Color (color score in whole apple), and W-General Evaluations (general evaluations score in whole apple). Among these, the deflection index showed the highest negative correlation to W-Shape (r = −0.78), highlighting the negative effects of the deflection index for the apple shape. F-Sourness (sourness score in fresh-cut apple) correlated positively with TA (r = 0.64), sucrose content (r = 0.76), and malic acid content (r = 0.63), indicating the good effect of sucrose and acidity on the overall taste of the apple. The results also showed that J-Taste (taste score in apple juice) correlated positively with TA (r = 0.65), and correlated negatively with TSS/TA (r = −0.68). This finding was in line with the previous study by Mureșan et al. [33], as well as our finding in Section 3.2.1.

In addition, HCA was conducted to manage large data to observe the degree of similarity among different apple cultivars and achieve quality classification. The fifteen studied apple cultivars were divided into three distinct clusters, with a relative distance of 8. Cluster I, the largest cluster, consisted of Aziteke, Bakeai, Magic Flute, Royal Gala, Red General, Red Delicious, and Zhongqiuwang. These cultivars stood out from the others due to their appropriate size, symmetrical shape, rich aroma volatile compounds, and overall evaluation. The high quality of the first visual and olfactory assessment made them suitable for direct consumption. While Aziteke had an appealing appearance, other factors (the lowest content of aroma volatile compounds (8821.36 μg/kg) and medium-scoring of general evaluation in sensory (21.3)) might prevent consumers from purchasing it repeatedly. Cluster II included the Fuburuisi, Sinike, Honglu, and Huashuo. The first three cultivars were characterized by high sugar content and higher acid content but unattractive appearance, which were suitable for the production of apple juice, cider, and cider vinegar. The juice from Huashuo had a special clean and light palate due to its low TSS value. Cluster III included the Miqila, Honey Crisp, Shandong Fuji, and Yanfu 3 cultivars. While they had some weaknesses such as less vibrant peel color and high deflection index values, these cultivars were favored by assessors due to their good flavor. Fresh-cut apples can compensate for any lack of shape and color, and a popular brand can shift consumers’ attention from appearance to flavor.

In summary, the study identified specific clusters of apple cultivars based on their suitability for direct consumption, apple juice production, and fresh-cut applications. The above results were in line with the chemical attributes measured in our samples, indicating that differences in sensory attributes and chemical profiles among apple cultivars were closely related and can be effectively evaluated by multivariate data analysis methods. Such findings shed light on the interrelationships between various quality parameters and sensory attributes of apple cultivars. More to the point, these findings revealed distinct clusters of apple varieties based on their quality and flavor indicators and could guide the selection of appropriate cultivars for different purposes, such as direct consumption or juice production.

## 4. Conclusions

The study evaluated the quality of various apple cultivars based on both physicochemical parameters and sensory attributes. The results revealed that Huashuo and Miqila exhibited higher regular globe shape indexes close to 1. However, Honglu, Honey Crisp, Shandong Fuji, Sinike, and Fuburuisi displayed less aesthetically pleasing appearances due to their higher deflection indexes, *L**, *b** values, and lower *a** values. Most of the apples had a sweet taste (TSS > 12 ^◦^Brix) and distinct sourness (TA > 0.08%). Fuburuisi, Sinike, and Aziteke were categorized as cultivars with high hardness. Royal Gala and Magic Flute were notable for their rich aroma.

Based on multivariate data analysis and sensory evaluation, all fifteen apple cultivars were grouped into three main clusters: direct consumption (Aziteke, Bakeai, Magic Flute, Royal Gala, Red General, Red Delicious, and Zhongqiuwang), fresh-cut apples (Miqila, Honey Crisp, Shandong Fuji, and Yanfu 3), and apple juice (Fuburuisi, Sinike, Honglu, and Huashuo). Cultivars deemed suitable for direct consumption exhibited favorable appearance, high sensory quality, and rich volatile compound aromas. Meanwhile, ideal cultivars for fresh-cut apples were characterized by excellent flavor despite having an unattractive appearance. Cultivars with high sugar and acid content were better suited for apple juice production, prioritizing flavor over appearance. The findings offer valuable insights that can assist the industry in selecting suitable cultivars for different apple products.

## Figures and Tables

**Figure 1 foods-12-03095-f001:**
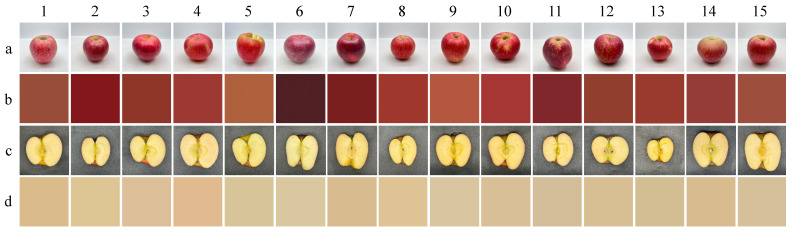
Morphological and color phenotype observation of fifteen apple cultivars. (**a**) Appearance of entire apples; (**b**) Fitting graphs of apple peel; (**c**) Color phenotype of apple flesh; (**d**) Fitting graphs of apple flesh. Numbers represent the cultivars: 1 (Aziteke); 2 (Bakeai); 3 (Fuburuisi); 4 (Red General); 5 (Honglu); 6 (Red Delicious); 7 (Huashuo); 8 (Royal Gala); 9 (Miqila); 10 (Honey Crisp); 11(Magic Flute); 12 (Shandong Fuji); 13 (Sinike); 14 (Yanfu 3); 15 (Zhongqiuwang).

**Figure 2 foods-12-03095-f002:**
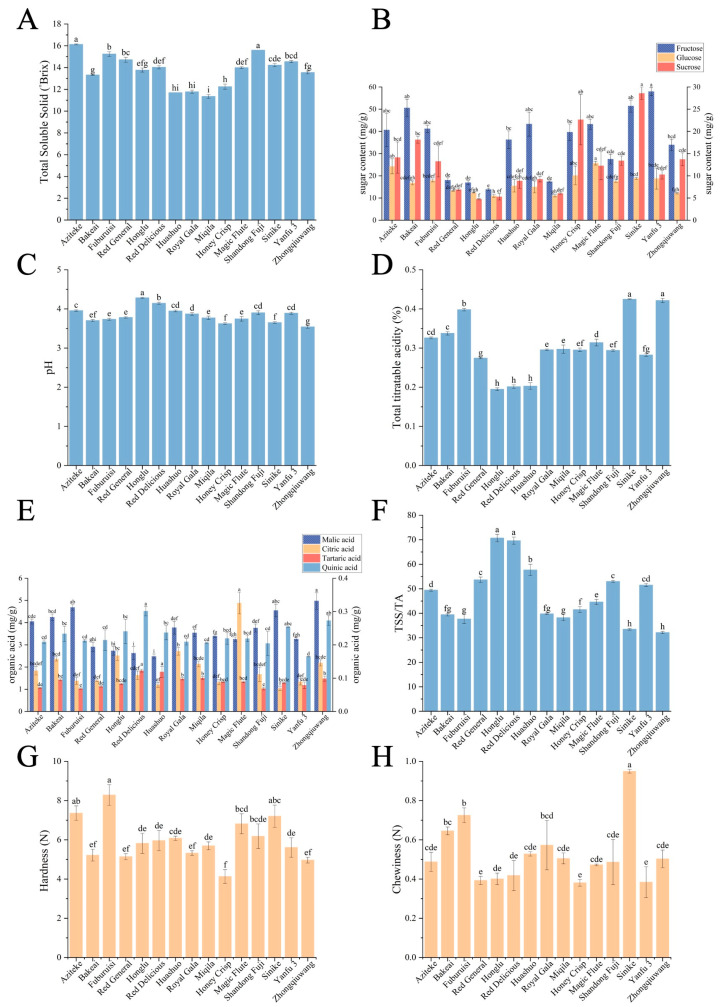
Comparison of internal qualities in fifteen apple cultivars. (**A**) total soluble solid (TSS), (**B**) sugars contents (fructose content base on left *y*-axis), (**C**) pH, (**D**) total titratable acidity (TA), (**E**) TSS/TA ratio, (**F**) organic acids contents (malic acid content base on the left *y*-axis), (**G**) hardness, (**H**) chewiness. The bars with different lowercase letters represent significant differences, *p* < 0.05.

**Figure 3 foods-12-03095-f003:**
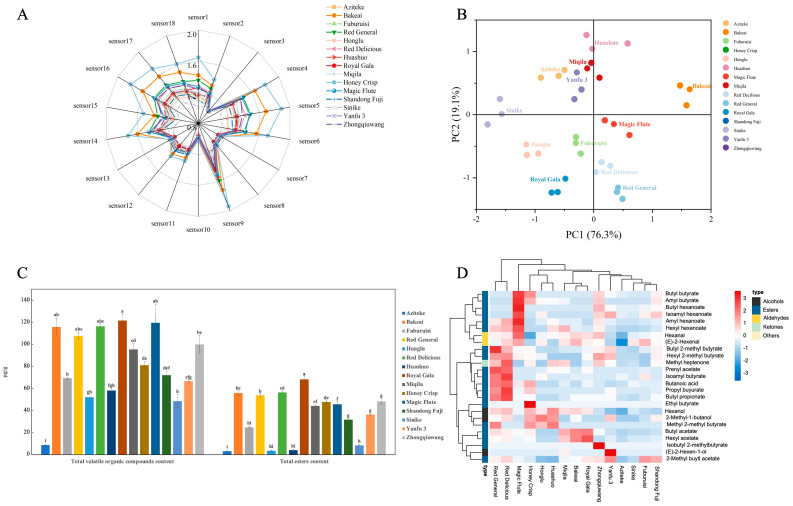
Comparison of aroma qualities in fifteen apple cultivars. (**A**) Radar chart of electronic nose data, (**B**) PCA score plot of electronic nose data, (**C**) total of volatile organic compounds and esters content, and (**D**) heat map analysis. The bars with different lowercase letters represent significant differences, *p* < 0.05.

**Figure 4 foods-12-03095-f004:**
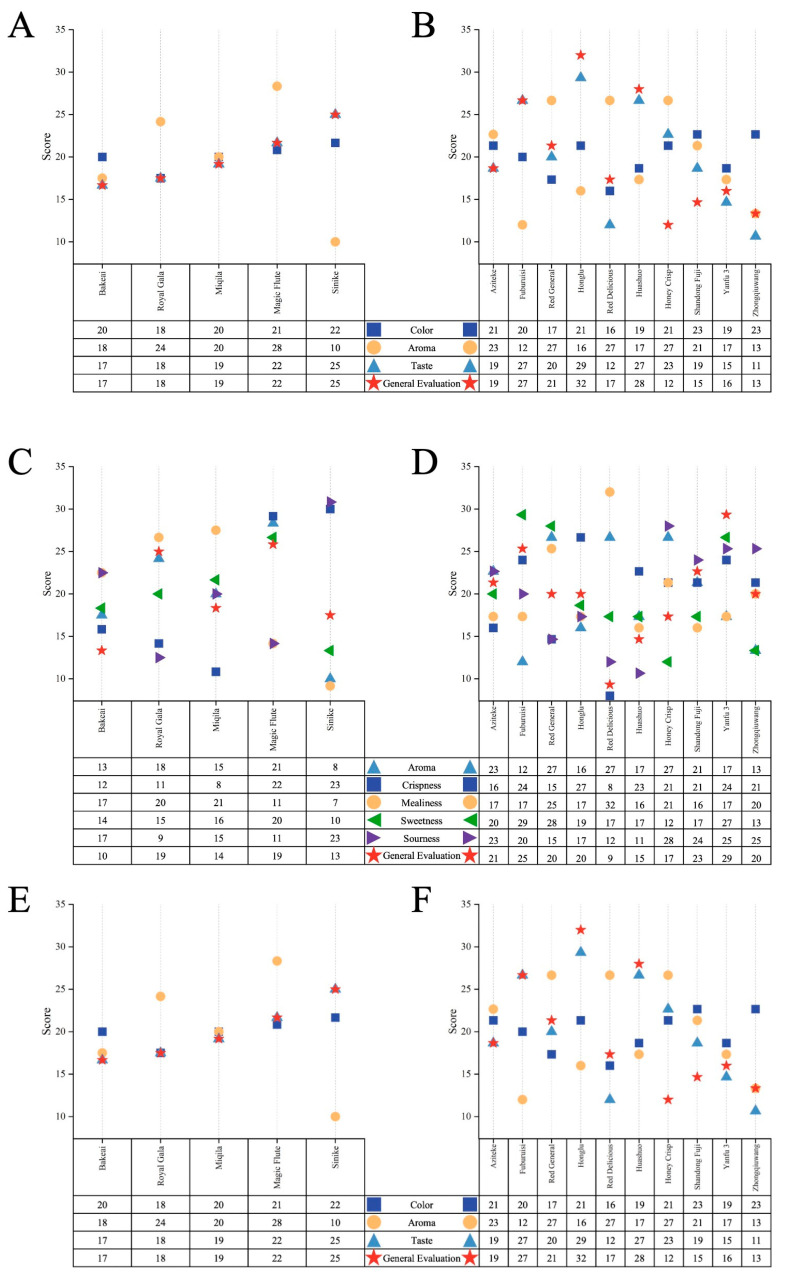
Comparison of sensory evaluation in fifteen apple cultivars. Scores of (**A**,**C**,**E**) small-size cultivars and (**B**,**D**,**F**) large-size cultivars for the whole apples, fresh-cut apples, and apple juice, respectively. Scores were expressed as the sum of measurement rank scores.

**Figure 5 foods-12-03095-f005:**
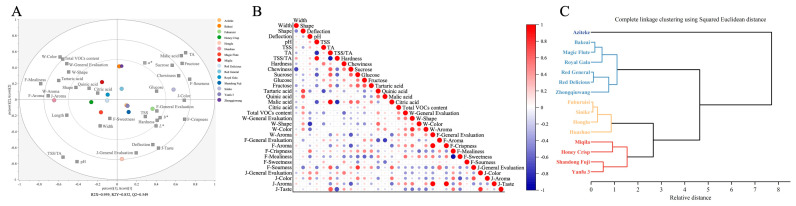
Chemometrics analysis of fifteen apple cultivars. (**A**) hierarchical cluster analysis, (**B**) PLS-DA loading plot showing relationships between sensory attributes and chemical profiles of apple cultivars (R2X = 0.959, R2Y = 0.832, Q2 = 0.549), and (**C**) Pearson correlation analysis. Circles represent a positive (red) or negative (blue) correlation between the quality indicators. The size of the circle represents the levels of the correlation coefficient, i.e., bigger areas indicate a higher correlation.

**Table 1 foods-12-03095-t001:** Appearance qualities in fifteen apple cultivars.

Cultivar	Length (mm)	Width (mm)	Shape Index	Deflection Index	Color Attributes (Apple Peel)			Color Attributes (Apple Pulp)		
					*L**	*a**	*b**	*L**	*a**	*b**
Aziteke	68.20 ± 2.56 ^fg^	78.90 ± 3.22 ^cd^	0.87 ± 0.03 ^ef^	0.15 ± 0.08 ^bc^	39.61 ± 3.11 ^bc^	21.35 ± 1.63 ^gh^	19.70 ± 1.41 ^de^	76.65 ± 0.51 ^cd^	1.00 ± 0.11 ^cd^	26.58 ± 0.77 ^a^
Bakeai	64.68 ± 2.49 ^gh^	66.93 ± 0.59 ^gh^	0.97 ± 0.04 ^bcd^	0.12 ± 0.04 ^bc^	25.16 ± 1.26 ^fg^	36.06 ± 0.74 ^a^	19.25 ± 1.83 ^de^	79.42 ± 0.17 ^a^	−0.75 ± 0.14 ^gh^	25.81 ± 0.44 ^ab^
Fuburuisi	71.64 ± 2.49 ^def^	78.02 ± 1.84 ^de^	0.92 ± 0.04 ^cde^	0.22 ± 0.10 ^abc^	32.97 ± 4.13 ^de^	30.82 ± 0.69 ^bc^	20.08 ± 1.90 ^cde^	77.79 ± 0.22 ^b^	2.00 ± 0.26 ^b^	21.34 ± 0.24 ^defg^
Red General	71.04 ± 1.79 ^ef^	80.56 ± 2.44 ^cd^	0.89 ± 0.01 ^def^	0.14 ± 0.06 ^bc^	35.67 ± 1.44 ^bcd^	31.21 ± 0.64 ^bc^	20.13 ± 0.74 ^cde^	76.92 ± 0.17 ^bcd^	3.93 ± 0.31 ^a^	25.29 ± 0.43 ^ab^
Honglu	76.66 ± 1.88 ^bcd^	80.39 ± 2.47 ^cd^	0.95 ± 0.03 ^bcd^	0.34 ± 0.07 ^a^	47.17 ± 3.43 ^a^	20.23 ± 4.61 ^h^	28.63 ± 2.27 ^a^	79.3 ± 0.35 ^a^	−1.45 ± 0.09 ^h^	22.71 ± 0.79 ^cde^
Red Delicious	81.58 ± 1.91 ^ab^	77.88 ± 1.07 ^de^	1.07 ± 0.03 ^a^	0.12 ± 0.04 ^bc^	19.55 ± 0.88 ^g^	19.59 ± 1.29 ^h^	5.79 ± 0.70 ^g^	79.77 ± 0.16 ^a^	−1.15 ± 0.09 ^gh^	21.06 ± 0.30 ^efg^
Huashuo	79.01 ± 3.89 ^abc^	77.88 ± 0.90 ^de^	1.01 ± 0.06 ^ab^	0.13 ± 0.06 ^bc^	23.06 ± 3.12 ^fg^	30.42 ± 1.31 ^bcd^	16.86 ± 1.90 ^ef^	75.47 ± 0.35 ^e^	1.55 ± 0.68 ^bc^	22.42 ± 0.45 ^de^
Royal Gala	63.04 ± 1.95 ^gh^	66.10 ± 1.48 ^hi^	0.96 ± 0.05 ^bcd^	0.13 ± 0.03 ^bc^	35.66 ± 0.83 ^bcd^	32.97 ± 0.99 ^ab^	23.79 ± 1.42 ^bc^	79.66 ± 0.30 ^a^	0.13 ± 0.12 ^ef^	25.85 ± 0.39 ^ab^
Miqila	73.56 ± 2.08 ^de^	74.47 ± 1.66 ^ef^	0.99 ± 0.05 ^abc^	0.09 ± 0.03 ^c^	47.04 ± 1.61 ^a^	25.28 ± 1.11 ^efg^	27.34 ± 1.57 ^ab^	80.02 ± 0.57 ^a^	−0.97 ± 0.27 ^gh^	19.97 ± 1.43 ^g^
Honey Crisp	71.18 ± 2.67 ^ef^	87.93 ± 1.74 ^a^	0.82 ± 0.04 ^f^	0.20 ± 0.08 ^abc^	36.13 ± 2.26 ^bcd^	37.36 ± 0.20 ^a^	22.14 ± 0.89 ^cd^	77.65 ± 0.19 ^b^	0.61 ± 0.19 ^de^	21.33 ± 0.87 ^defg^
Magic Flute	75.96 ± 3.17 ^cde^	70.73 ± 2.44 ^fg^	1.07 ± 0.03 ^a^	0.17 ± 0.05 ^bc^	27.00 ± 2.85 ^ef^	26.94 ± 0.87 ^cdef^	13.59 ± 1.22 ^f^	77.26 ± 0.20 ^bcd^	0.79 ± 0.17 ^cde^	20.21 ± 0.44 ^fg^
Shandong Fuji	70.76 ± 4.17 ^ef^	85.37 ± 1.62 ^ab^	0.83 ± 0.05 ^f^	0.27 ± 0.10 ^ab^	33.37 ± 4.38 ^cde^	25.85 ± 2.06 ^defg^	19.30 ± 1.38 ^de^	77.34 ± 0.13 ^bcd^	0.91 ± 0.24 ^cde^	23.01 ± 0.82 ^cd^
Sinike	60.33 ± 2.36 ^h^	62.53 ± 0.18 ^i^	0.96 ± 0.04 ^bcd^	0.25 ± 0.08 ^abc^	36.05 ± 2.08 ^bcd^	30.45 ± 1.86 ^bcd^	19.90 ± 1.54 ^de^	76.47 ± 0.65 ^d^	−0.81 ± 0.18 ^gh^	22.03 ± 1.40 d^ef^
Yanfu 3	71.28 ± 2.48 ^ef^	81.09 ± 2.90 ^cd^	0.87 ± 0.03 ^ef^	0.10 ± 0.03 ^c^	34.89 ± 1.17 ^bcd^	28.91 ± 0.22 ^bcde^	17.10 ± 1.39 ^ef^	76.94 ± 0.38 ^bcd^	1.34 ± 0.38 ^bcd^	24.53 ± 0.75 ^bc^
Zhongqiuwang	83.72 ± 1.67 ^a^	82.05 ± 1.21 ^bc^	1.02 ± 0.04 ^ab^	0.12 ± 0.03 ^bc^	40.33 ± 1.25 ^b^	23.94 ± 3.26 ^fgh^	21.25 ± 1.25 ^cd^	77.53 ± 0.61 ^bc^	−0.59 ± 0.70 ^fg^	19.54 ± 0.36 ^g^

Data were shown as the mean ± standard deviation. Different letters in the same column show significant differences at *p* < 0.05 determined by Tukey’s test.

**Table 2 foods-12-03095-t002:** The information of discriminant volatile compounds of fifteen apple cultivars.

Compounds ^a^	CAS	Odor Description ^b^	RI ^c^	RI* ^d^
Alcohols				
2-Methyl-1-butanol	137-32-6	ethereal fusel alcoholic fatty greasy winey whiskey leathery cocoa	741	736
Pentanol	71-41-0	fuel oil sweet balsam	767	764
(E)-2-Hexen-1-ol	928-95-0	fresh green leafy fruity unripe banana	857	858
Hexanol	111-27-3	ethereal fuel oil fruity alcoholic sweet green	874	874
Hydroxy	111-70-6	musty leafy violet herbal green sweet woody peony	972	972
2-Ethyl-1-hexanol	104-76-7	citrus fresh floral oily sweet	1030	1029
Octanol	111-87-5	waxy green orange aldehydic rose mushroom	1072	1070
Nonanol	143-08-8	fresh clean fatty floral rose orange dusty wet oily	1172	1172
Esters				
Propyl acetate	109-60-4	solvent celery fruity fusel raspberry pear	713	717
Butanoic acid	623-42-7	fruity apple sweet banana pineapple	723	721
Methyl 2-methyl butyrate	868-57-5	ethereal estery fruity tutti frutti green apple lily of the valley powdery fatty	775	770
Ethyl butyrate	105-54-4	fruity juicy fruit pineapple cognac	776	776
Butyl acetate	123-86-4	ethereal solvent fruity banana	815	813
Ethyl 2-methyl butanoate	7452-79-1	sharp sweet green apple fruity	892	893
Propyl butyrate	105-66-8	fruity sweet apricot pineapple rancid sweaty	898	900
Butyl propionate	590-01-2	earthy sweet weak rose	910	910
Prenyl acetate	1191-16-8	sweet fresh banana fruity jasmin ripe heliotrope balsam	921	923
Methyl hexanoate	106-70-7	ethereal fruity pineapple apricot strawberry tropical fruit banana bacon	926	938
Butyl butyrate	109-21-7	fruity banana pineapple green cherry tropical fruit ripe fruit juicy fruit	996	993
Isobutyl 2-methylbutyrate	2445-67-2	sweet fruity	1002	1002
Leaf acetate	3681-71-8	fresh green sweet fruity banana apple grassy	1004	1005
Hexyl acetate	142-92-7	fruity green apple banana sweet	1013	1012
Butyl 2-methyl butyrate	15706-73-7	fruity tropical green ethereal herbal celery cocoa jammy peach grassy	1044	1047
Isoamyl butyrate	106-27-4	fruity green apricot pear banana	1061	1059
Amyl butyrate	540-18-1	sweet fruity banana pineapple cherry tropical	1094	1091
Heptyl acetate	112-06-1	fresh green rum ripe fruit pear apricot woody	1111	1110
Benzyl acetate	140-11-4	sweet floral fruity jasmin fresh	1167	1172
Butyl hexanoate	626-82-4	fruity pineapple berry apple juicy green winey waxy cognac soapy	1191	1189
Hexyl 2-methyl butyrate	10032-15-2	green waxy fruity apple spicy tropical	1237	1237
Isoamyl hexanoate	2198-61-0	fruity banana apple pineapple green	1253	1254
Amyl hexanoate	540-07-8	sweet green fruity estry pineapple apple pear fatty	1288	1289
Hexyl hexanoate	6378-65-0	herbal fresh cut grass vegetable fruity	1386	1387
Aldehydes				
Hexanal	66-25-1	fresh green fatty aldehydic grass leafy fruity sweaty	802	802
(E)-2-Hexenal	6728-26-3	green banana aldehydic fatty cheesy	850	850
Heptanal	111-71-7	fresh aldehydic fatty green herbal wine-lee ozone	903	903
(E, E)-2,4-Hexadiena	142-83-6	sweet green spicy floral citrus	912	914
Benzaldehyde	100-52-7	strong sharp sweet bitter almond cherry	964	964
5-Methyl furfural	620-02-0	spice caramel maple	964	969
Octanal	124-13-0	aldehydic waxy citrus orange peel green herbal fresh fatty	1003	1004
Phenyl acetaldehyde	122-78-1	green sweet floral hyacinth clover honey cocoa	1047	1046
(E)-2-Nonenal	18829-56-6	fatty green cucumber aldehydic citrus	1062	1065
(E)-2-Octena	2548-87-0	fresh cucumber fatty green herbal banana waxy green leaf	1061	1062
Decanal	112-31-2	sweet aldehydic waxy orange peel citrus floral	1026	1024
(E)-2-Octena	2548-87-0	fresh cucumber fatty green herbal banana waxy green leaf	1061	1062
Ketones				
Methyl heptenone	110-93-0	citrus green musty lemongrass apple	986	986
2-Nonanone	821-55-6	fresh sweet green weedy earthy herbal	1091	1091
Geranyl acetone	689-67-8	fresh rose leaf floral green magnolia aldehydic fruity	1451	1456
Others				
Styrene	100-42-5	sweet balsam floral plastic	890	895
(Z)-2-Heptenal	57266-86-1	grass	960	963
p-isopropyl toluene	99-87-6	fresh citrus terpene woody spice	1029	1028
Estragole	140-67-0	sweet sassafrass anise spice green herbal fennel	1199	1199
Alpha-curcumene	644-30-4	herbal	1486	1486
(E)-Beta-farnesene	18794-84-8	woody citrus herbal sweet	1456	1456
(Z, E)-Alpha-farnesene	26560-14-5	NF	1491	1489
Alpha-farnesene	502-61-4	citrus herbal lavender bergamot myrrh neroli green	1507	1506
Beta-bisabolene	495-61-4	balsamic woody	1512	1510

“NF”: not found. ^a^: The reliability of the identification proposal is carried out: mass spectrum and retention index agreed with database or literature. Identification methods: mass spectrometry and retention indices. ^b^: Odor descriptions were obtained from literature data (https://www.thegoodscentscompany.com (accessed on 22 October 2022). ^c^: Calculated retention index (RI) on DB-5 MS column. ^d^: The retention indexes (RI*) of the references on the DB-5MS column were obtained from literature data (https://webbook.nist.gov/chemistry/cas-ser/ (accessed on 8 October 2022).

## Data Availability

Data is contained within the article.

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
