# Peer review of "Chemometric Classification of Apple Cultivars Based on Physicochemical Properties: Raw Material Selection for Processing Applications"

_foods, 2023, doi:10.3390/foods12163095_

Round 1

Reviewer 1 Report

The article entitled "Chemometric comparison and classification of different apple cultivars based on physicochemical quality properties: Raw material selection for various processing applications" presents a quite serious and rigorous study. The authors have done a great job. The authors have studied the the physicochemical parameters and sensory attributes differences among fifteen apple cultivars and identify characteristic qualities suitable for various processed apple products using chemometric analysis. These results are very important for producers. However, the authors should clarify some points of the paper as follows:

Section 2.2 and section 2.3. I suggest combining them, because both cections describe methods of sample preparation.

Section 2.4. Fruit phenotypes. Lines 101 to 104 do not describe the determination of phenotypicity, but the methodology of color coordinate research. In addition, the CIELab color coordinate methodology should be described in more detail in the next section.

Lines 114-118. I doubt if the Texture Analyzer TA.TX Plus is made in China. As far as I know it is made in the UK.

Section 2.6. Hardness methodology is described, but chwiness is not. These are completely different research methods.

Lines 179, 203. p must be Italic, p

It is not clear how the chewiness results in Figure 2 H were obtained.

In Figure 2, hardness should be expressed in SI units, i.e. N/m2

It is not clear why Total volatile organic compounds content (Figure 3D) is expressed in mkg/kg and not in mg/g as in other Fig.

For all these reasons, I consider that the article needs a minor revision.

Reviewer 2 Report

The authors in this manuscript present information about quality of various apple cultivars based on their chemical and sensory features. Manuscript in general is well structured and written, but some questions could be clarified and improved. In my opinion this manuscript need some revision before considering for publication.

My general comments are presented below

Title is too long and could be shortened.

Too many digits in data. Round your results taking into account uncertainty of measurements.

In introduction complete the information what characteristics of apples predispose to what use (for example for direct consumption desirable is good appearance).

3.3. Apple aroma – add some information on desirable and undesirable aromatic compounds in apples.

Conclusions should be more reflected in the research results.

How many apples were used for each measurement?

Specific comments are listed below:

Line 18 Expand abbreviations where it first appeared.

Line 33 Could you check it is Malus pumila or Malus domestica?

Line 71 – 78 Apple cultivars were cultivate in one orchard? Were they collected at the same time? How many apples did one sample/cultivar represent? Provide more details.

Line 101 – 104 Parameters during color determination is very important. Provide more details. How many apples/points were measure? It was mean from replications? Which point on apple surface were selected? On apple skin blush is visible, how did you cope with that? Light source?

Line 114 – 118 What does mean (how was calculated) hardness and chewiness? Apples were puncture on skin or flesh? In the middle or close to stalk?

Line 193 What was the criteria for the classification into small and big apples?

Line 246 – 247 In my opinion content of sugars and acidity are not teste, but chemical features.

Line 256 – 258 Figure 2B Contents of sugars are very low (compare with Aprea E, Charles M, Endrizzi I, Laura Corollaro M, Betta E, Biasioli F, Gasperi F. Sweet taste in apple: the role of sorbitol, individual sugars, organic acids and volatile compounds. Sci Rep. 2017 Mar 21;7:44950. doi: 10.1038/srep44950. PMID: 28322320; PMCID: PMC5359574.) How could you explain that?

Line 261 For me these parameters are not “taste qualities”.

Line 317, Figure 3C This figure is unnecessary and misleading. Is it the number of compounds or their content? A few compounds can make up a large proportion of the total content.

Line 324 You refer to Figure 2B?

Line 367 Sensory evaluation ranking test from 1 to 5, but results higher scores?

Line 387 Maybe presents sensory results in tables. The charts are not readable enough.

Line 393 PCA analysis, at Figure 5A you presented PCA plot, two principal components were identified, which explain lower than 50% of the variability. I think you should check this analysis was conducted properly.

Line 442 – 444 Crispness is texture, not taste parameter. I think you should correlate sugars contents with sensory analysis.

Reviewer 3 Report

1- Supplementary file for HPLC & GC should be made in a table form

2- Mention the commercial use of fresh-cut apple and its significance in market.

3- Differences between cNose & eNose should be also clarified

4- Line 18 (TA, TSS/TA) should be written in full in abstract 

5- Line 152 3mm cubes should be written mm3

minor English editing is needed
